# Effect of Si Contents on the Properties of Ti15Mo7ZrxSi Alloys

**DOI:** 10.3390/ma16144906

**Published:** 2023-07-09

**Authors:** Cristina Jimenez-Marcos, Julia Claudia Mirza-Rosca, Madalina Simona Baltatu, Petrica Vizureanu

**Affiliations:** 1Mechanical Engineering Department, Las Palmas de Gran Canaria University, 35017 Tafira, Spain; cristina.jimenez112@alu.ulpgc.es; 2Materials Engineering and Welding Department, Transilvania University of Brasov, 500036 Brasov, Romania; 3Department of Technologies and Equipments for Materials Processing, Faculty of Materials Science and Engineering, Gheorghe Asachi Technical University of Iaşi, Blvd. Mangeron, No. 51, 700050 Iasi, Romania; peviz@tuiasi.ro; 4Technical Sciences Academy of Romania, Dacia Blvd 26, 030167 Bucharest, Romania

**Keywords:** titanium-based alloys, microstructure, corrosion behavior, mechanical properties

## Abstract

The main purpose of this research is to evaluate the mechanical characteristics and biocompatibility of two novel titanium alloys, Ti15Mo7ZrxSi (x = 0, 0.5, 0.75, 1). These samples had already undergone grinding, polishing, cutting, and chipping. Electrochemical, metallographic, three-point bending, and microhardness studies were conducted on the studied materials to determine their corrosion behavior, microstructure, Young’s modulus, and hardness. The first investigations revealed that both samples had biphasic and dendritic structures, elastic moduli that were between the highest and minimum values achieved by around 20 GPa, and favorable behavior when in contact with physiological fluids at ambient temperature. Ti15Mo7Zr0.5Si and Ti15Mo7Zr0.75Si, the research samples, had greater corrosion potentials, reduced corrosion rates, and therefore higher corrosion resistance, as well as modulus of elasticity values that were comparable to and closer to those of human bone. The results of this investigation indicate that both alloys exhibit favorable corrosion behavior, great biocompatibility, Young’s modulus results lower than those of conventional alloys used in biomedical implants, and hardness values higher than commercially pure titanium.

## 1. Introduction

Nowadays, there is a growing interest in the development of biomaterials for various medical applications, such as orthopedics [1], drug administration [2,3], dentistry [4], tissue engineering [5], and cardiovascular systems [6]. These biomaterials aim to improve the quality of life and longevity of individuals, particularly in the context of chronic musculoskeletal diseases like osteoarthritis, which affects a significant portion of the population.

Biocompatibility is a crucial requirement for any biomaterial intended for medical use. Biocompatibility refers to the ability of a material to fulfill its function in medical treatment without causing adverse reactions in the body. To be successful, biomaterials must possess specific properties, including high ductility, fatigue and wear resistance, absence of cytotoxicity, a combination of high strength and low Young’s modulus similar to human cortical bone (ranging from 10 to 30 GPa), and the ability to integrate well with the surrounding bone.

While polymers and ceramics are commonly used biomaterials [7], they often exhibit low mechanical strength and brittleness, limiting their applications in harsh conditions. Therefore, metallic materials, particularly titanium and its alloys [8], have gained prominence in orthopedic surgery. Approximately 70–80% of implants are made from metallic biomaterials, such as 316L stainless steel, CoCrMo alloys, Ti6Al4V, and NiTi alloys [9].

However, using metals as implants presents challenges as the higher elastic modulus of metals compared to bone can lead to stress shielding, causing osteoporosis due to the mismatch in mechanical properties. Moreover, many metallic biomaterials release toxic ions into adjacent tissues, exhibit poor wear and corrosion resistance, and may have low biocompatibility [10]. To address these issues, the addition of alloying components that do not have harmful effects on the body has been explored [11,12].

Corrosion is a significant concern when using metallic implants, as it can lead to material deterioration and subsequent implant failure. To prevent corrosion, a protective film is usually formed on the material’s surface to make it impermeable to the surrounding environment. Titanium and its alloys have emerged as suitable biomaterials for various applications due to their biocompatibility, corrosion resistance, mechanical properties, low modulus of elasticity, and thermal stability [13,14,15,16]. Titanium’s ability to form a stable oxide coating on its surface contributes to its excellent corrosion resistance.

Different types of titanium-based alloys exist, including α, α + β, and β titanium alloys. The addition of alloying elements allows for the modification of alloy properties. α-stabilizers (C, N, O, Al) increase the allotropic transformation temperature, while β-stabilizers (V, Nb, Mo, Ta, Fe, Mn, Cr, Co, W, Ni, Cu, Si) decrease it [8,15,17].

Titanium–molybdenum (Ti-Mo) alloys have gained attention as potential biomaterials due to their favorable mechanical and corrosion resistance. Ti-Mo alloys can form a stable oxide layer on their surface, protecting them from corrosion and degradation in aggressive environments [18,19,20].

The design considerations for Ti-Mo-Zr alloys were based on several factors influenced by fabrication method [21] or another alloying element, such as chromium [22]. Titanium is considered to have extremely low toxicity for the human body and exhibits excellent interaction with bone due to its corrosion resistance and lack of rejection by the body. Molybdenum is a beta-stabilizing element with low toxicity and can help adjust the alloy’s mechanical properties to be more similar to human bone by decreasing the modulus of elasticity. Zirconium, a neutral stabilizer, is becoming increasingly desirable for medical applications due to its biocompatibility, low modulus of elasticity, and corrosion resistance [23].

In recent years, Ti-Mo-Zr-Si alloys have been developed as potential biomaterials by incorporating small quantities of silicon into Ti-Mo-Zr alloys [24]. Silicon is a biocompatible element found in human bone, and its addition to the alloy improves corrosion resistance, creep resistance at high temperatures, ductility, and strength. It also contributes to reducing the elastic modulus, making it closer to that of human bone [25,26]. The specific effects of silicon content on titanium alloys can vary depending on the other alloying elements, processing conditions, and the intended application of the alloy. Therefore, comprehensive testing and characterization are necessary to determine the optimal silicon content for a particular titanium alloy to achieve the desired properties.

In the current study, a Ti-15Mo-7Zr base alloy is considered, and we analyze the effect of the addition of silicon (with 0%, 0.50%, 0.75%, and 1.00% Si) in the microstructure, and the corrosion resistance, microhardness, and elastic modulus of the alloys obtained in an effort to develop new alloys for biomedical applications.

## 2. Materials and Methods

### 2.1. Material Preparation

The examined alloys’ chemical compositions (%wt) are shown in Table 1. The alloys were produced using the arc remelting method in an argon environment. The melting process takes place under high vacuum conditions, achieved by utilizing a pump system to evacuate the working chamber. When the samples are melted, the presence of oxygen leads to a significant decrease in vacuum within the working chamber. Once the high vacuum is restored, argon is introduced into the working chamber, where the samples are melted, under atmospheric conditions. This approach ensures the proper temperature and facilitates easy handling of the samples, resulting in a high level of uniformity in their chemical composition.

The Faculty of Materials Science and Engineering at Gheorghe Asachi Technical University in Iasi, Romania, carried out this technique. The alloys were melted, then remelted six times (three times in each face), and finally formed into an ingot in order to obtain the proper homogeneity. Moreover, for their preparation and testing, Las Palmas de Gran Canaria University (Las Palmas de Gran Canaria, Spain) received a portion of the ingots.

Several operations were carried out as a first step to analyze the electrochemical, metallographic, bending, and microhardness properties of the studied samples, such as embedding the samples by adding epoxy resin or catalyst in a 4:1 ratio into a mold, which was demolded after 24 h. The specimens were then cut into 1 to 1.5 mm thick plates using a grinding wheel and jaws on a Buehler IsoMet 4000 precision saw (Chicago, IL, USA). For the mechanical and electrochemical testing, the sliced specimens were reassembled. The specimens were then polished using the Struers TegraPol-11 polishing machine (Copenhagen, Denmark) at a force of 20 N and at a speed of 300 rpm. The progressive carbide grinding method was utilized. Carbide abrasive sheets of increasing grit were utilized, beginning with 400 grits, and ending with P2500 grit. Finally, mirror polishing cloths with 0.1 μm α alumina suspension were utilized. These procedures for sample preparation for metallographic testing followed ASTM E3-11(2017) [27]. In order to realize the mechanical tests, vertical slices with a thickness of about 2 mm were produced once more on the cutting machine.

### 2.2. Microstructure

The compounds and phases that make up a metallic substance are arranged spatially in metallography, together with any impurities or potential mechanical faults.

Images of the surfaces of every specimen were acquired using the Axio Vert.A1 MAT ZEISS optical metallographic microscope (Jena, Germany) to analyze the microstructure. At intervals of about 15 s, each sample was submerged in Kroll’s reagent, which is made of 20 mL glycerin, 30 mL hydrochloric acid, and 10 mL nitric acid, and the attacked surface was photographed. The test was carried out three times.

### 2.3. Electrochemical Tests

A sample was placed in an electrochemical cell with three electrodes for the electrochemical tests: the samples acted as the working electrodes, a saturated calomel electrode served as the reference electrode, and a platinum electrode acted as the counter electrode. The area of each sample was determined to run the tests. The mmol/L values of the Grifols Laboratories’ (Barcelona, Spain) Ringer solution were as follows: Na^+^ 129.9, Cl^−^ 111.7, C_3_H_5_O_3_ 27.2, K^+^ 5.4, and Ca^2+^ 1.8.

Corrosion Potential, Corrosion Rate, and Electrochemical Impedance Spectroscopy were performed by applying the BioLogic Essential SP-150 potentiostat (Seyssinet-Pariset, France). The tests were performed at 25 °C in aerated Ringer solution.

#### 2.3.1. Corrosion Potential (E_corr_)

Applying the “E_corr_ vs. Time” approach found in the Ec-Lab program, the 24 h corrosion potential of each sample was determined. Potential readings were taken every 300 s or every time there was a 100 mV change in potential. The collected data were analyzed, and a potential versus time graph was created.

#### 2.3.2. Corrosion Rate (V_corr_)

The “Linear Polarization” approach was selected to carry out these experiments [28], and the sample surface area value and the 20 min test period were entered to verify its viability. With data taken every 0.50 s, the potential scanning revealed a 0.167 mV/s time-variation relationship from −0.025 to 0.025 V against open circuit potential (OCP) and intensity maintained at 100 % during the potential scanning [29]. Following the presentation of these linear polarization curves, EC-Lab’s “Tafel Fit” approach was used to obtain the corrosion rate estimates for each sample.

#### 2.3.3. Electrochemical Impedance Spectroscopy (EIS)

The AC impedance measurements were recorded with an AC potential amplitude of 20 mV and single sine wave measurements were conducted at frequencies between 10^−1^ and 2 × 10^−5^ Hz. To analyze the characteristics of the oxide film, the impedance spectra were recorded at 7 different DC potentials around corrosion potential, in the range E_corr_ ± 300 mV, with a 100 mV step from E_corr_, permitting the system to stabilize for 5 min at each potential. To represent these data, Nyquist and Bode diagrams were utilized, and for their simulation, equivalent circuits (EC) were employed [30].

### 2.4. Three-Point Bending Test

The Bose ElectroForce^®^ 3100 machine (Framingham, MA, USA) was used to perform the three-point bending method; it complies with ISO 7438:2020 [31] and has a 20 N force resistance limit.

In order to realize this methodology, each rectangular cross-section specimen was positioned at the extremities of the bottom shank of the testing apparatus, with a distance between supports ranging from 7.80 to 10.63 mm depending on the specimen length. During the experiment, the specimen was loaded vertically while moving at a 3 mm/s linear speed at its center until it reached its yield stress or broke. The obtained values of the applied force against the displacement of the samples were plotted, and their slope was calculated to establish the modulus of elasticity.

### 2.5. Microhardness Test

Using the Future Tech FM-810 hardness tester (Kawasaki, Japan), for each sample’s applied load, in this case 5, 25, and 50 gf, 10 measurements were taken, in accordance with ISO 14577-1:2015 [32]. The mark may contain fragments of many phases as the stress rises, providing an approximation of the material’s total hardness. When relatively light weights are placed, it is likely that the mark will only be discovered in one phase, allowing the hardness of that phase to be assessed. The Vickers microhardness values were then computed automatically by the iVicky software (v2.0, Sinowon, Dongguan, China) using the observed diagonal lengths. The number of indents created was plotted against the scan length.

## 3. Results and Discussion

### 3.1. Microstructural Analysis

Figure 1 displays the surfaces of the four studied samples after they were etched with the reagent.

At room temperature, cpTi has a hexagonal closest-packed (HCP) structure with an α-phase, but beyond 883 °C, it has a body-centered cubic (BCC) structure with a β-phase [33]. 

In contrast to Zr, which is regarded as a neutral element because it virtually has no influence on the α/β-phases, the β-phase is stable at temperatures below 883 °C when Mo and Si are present [8].

After being chemically etched, it can be seen that all four of the tested samples had a biphasic and dendritic structure. Additionally, the addition of Si caused the interdendritic zone to expand while the size of the dendrites dropped by about 12% as the silicon concentration increased with 0.25%. Fine grains hinder the propagation of cracks, enhancing the material’s resistance to fatigue.

### 3.2. Electrochemical Tests

#### 3.2.1. Corrosion Potential (E_corr_)

A quantitative measure of corrosion behavior is corrosion potential evolution over time, although this information is still insufficient for a thorough examination.

Figure 2 and Table 2 show the curves of corrosion potential as a function of time after 24 h of immersion in Ringer’s solution with their respective values of initial potential, after 1 hour and after 24 h of testing. The potential, known as open-circuit potential (OCP) in these circumstances, reveals the sample’s tendency for corrosion. From the data obtained after one hour of immersion, the potential of the samples Ti15Mo7Zr, Ti15Mo7Zr0.75Si, and Ti15Mo7Zr1Si increases due to passivation of the surfaces, reaching values between −0.337 V and 0.328 V, while the potential of Ti15Mo7Zr0.5Si slightly decreases to −0.227 V.

Overall, during the 24 h experiment, the potentials of every studied sample increased because of the passive layer thickening, reaching final values of −0.194 V, −0.194 V, −0.263 V, and −0.308 V, respectively. Therefore, the potential curves tended to increase progressively until signs of a possible stabilization of the potential could be observed, suggesting that the passive layer is thermodynamically resistant under Ringer’s solution conditions, with the exception of Ti15Mo7Zr0.75Si, which presents a plot with many irregular peaks.

Since the corrosion potentials values of Ti15Mo7Zr and Ti15Mo7Zr0.5Si are equal and they are higher to those of the examined samples and cpTi (from −0.10 V to −0.15 V), this indicates that the inclusion of Si alters the passive layer’s properties.

As the Si content in the titanium alloy increases, the corrosion potential becomes more negative. This shift towards more negative values suggests that the alloy becomes more susceptible to corrosion and indicates a decrease in the alloy’s stability in the given environment. If the added silicon content exceeds 0.5%, the open circuit potential shifts to more negative values due to the formation of galvanic couples between the Si-containing regions and the rest of the titanium alloy, leading to localized electrochemical reactions and a negative shift in the OCP. The Si content may influence the composition, thickness, and stability of the passive oxide layer formed on the surface of the alloy. Changes in the passivation behavior can result in variations of the OCP.

#### 3.2.2. Corrosion Rate (V_corr_)

Figure 3 shows the results of the linear polarization technique, which was performed to measure the alloys’ rate of corrosion, displayed on a semi-logarithmic scale of the current results.

Ti15Mo7Zr1Si shows higher values of anodic corrosion potential and current (E_corr_ and I_corr_, respectively), which is a measure of how much the alloy has been oxidized, than the other samples studied, as can be seen in Table 3. By examining the curve displayed against open circuit potential (OCP) across a range of 0.25 V, Tafel slopes (β_c_ and β_a_) were calculated. An alloy prone to passivation will have a value of a greater than c, while an alloy prone to corrosion will have a value of a less than c. In our case, all four samples have a tendency to passivate, i.e., to have a passive layer created on their surface.

Moreover, using the corrosion current (I_corr_), the constant (K) that establishes the corrosion rate units (1.288 × 10^5^ miliinches/A-cm-year), the equivalent weight (EW) in g/eq, the density (d) in g/cm^3^, and the area (A) of each sample (in cm^2^), Table 3 displays the Tafel curve parameters and the corrosion rate (V_corr_) of the tested samples.
(1)$$V_{corr}=\frac{I_{corr}\cdot K\cdot EW}{d\cdot A}$$

In this case, a minimum V_corr_ of 0.03 μm/year (Ti15Mo7Zr0.5Si) and a maximum V_corr_ of 0.22 μm/year (Ti15Mo7Zr1Si) were achieved, whereas commercial pure titanium, under similar circumstances, had a greater V_corr_ of 0.12 μm/year.

As the polarization resistance (R_p_) value rises, the alloy’s corrosion resistance increases, according to the results of the “Rp Fit” study performed using the EC-Lab program on the linear polarization curves of the V_corr_. Ti15Mo7Zr0.5Si and Ti15Mo7Zr0.75 alloys are extremely corrosion-resistant, with the R_p_ even reaching 10^6^ Ω·cm^2^ for highly corrosion-resistant materials, whereas Ti15Mo7Zr and Ti15Mo7Zr1Si had lower values of about 10^5^ Ω·cm^2^. Thus, the Ti15Mo7Zr1Si sample obtained the worst corrosion resistance [34].

#### 3.2.3. Electrochemical Impedance Spectroscopy (EIS)

A high-throughput method for examining interfacial properties linked to processes occurring on the alloy surfaces is electrochemical impedance spectroscopy. EIS has a variety of benefits over other electrochemical techniques, since it is a steady-state methodology that can probe, in this case, from 0.1 Hz to 1 × 10^6^ Hz and can quantify very small signals.

Each alloy’s EIS data are shown in Nyquist and Bode plots for the seven distinct potentials that were recorded in Ringer solution. By measuring EIS at both cathodic and anodic potentials relative to the corrosion potential, it is possible to obtain a comprehensive understanding of the corrosion behavior of the alloy over a range of electrochemical conditions. This information helps to evaluate the overall corrosion resistance and stability of the alloy in the given environment and identify critical potentials or conditions that favor corrosion or passivation.

When using Nyquist graphs, the hypothetical impedance results were contrasted with the actual values. For each and every applied potential, a capacitive arc is shown in Figure 4. In general, in these Nyquist curves, a capacitive arc is shown and it can be seen that the impedance values increase as more positive potentials are applied.

The logarithm curves of the impedance modulus and phase shift angle of the Ti15Mo7ZrxSi samples dissolved in Ringer’s solution are displayed using Bode plots. 

Figure 5 and Figure 6 show the Bode impedance and Bode phase diagrams, respectively, while Table 4 shows the results obtained from E_corr_ and ±300 mV versus E_corr_.

Figure 5 and Table 4 indicate greater impedance values for the four samples at the lowest frequency (0.1 Hz), with lower values for the Ti15Mo7Zr0.75Si sample and higher results for the Ti15Mo7Zr0.5Si sample, showing that the latter alloy’s corrosion resistance has improved. In addition, the impedance values tended to increase the more positive the applied potentials were. In addition, a specific behavior of passive film growth was observed for each sample, which has a tendency to exhibit a capacitive behavior.

Figure 6 and Table 4 show the curves and maximum values obtained for the phase angles of the samples, which tend to increase the more positive the applied potentials are. As a result, after examining the studied samples, it was found that the Bode phase curves showed a single-phase process. Furthermore, with increasing potential value, the phase angle tends to increase and the process takes place in a single step, i.e., in a time constant, when the curve starts to decline, although in the curves of the Ti15Mo7Zr, Ti15Mo7Zr0.5Si, and Ti15Mo7Zr0.75Si samples there is the possibility of the onset of a second time constant.

The equivalent circuit model R(QR)(QR), which best fits the experimental findings of the study for Ti15Mo7ZrxSi alloys, is shown in Figure 7. This circuit shows that, up until the alloy is achieved, the surface of the samples exhibits a resistance to dissolving, together with porous and compact passive films.

The resistance in ohms of the electrolyte was represented in the model and in Equation (1) as R_sol_, the resistances of the porous and compact passive films as R_1_ and R_2_, respectively, and the capacitances of the passive film as CPE_1_ and CPE_2_, which are the constant phase elements. The constant phase element reproduces a capacitor (n = 1.0), a semi-infinite Warburg impedance (n = 0.5), or a resistor (n = 0.0) as a function of the values of the parameters n_1_ and n_2_, as well as the applied frequency, f.
(2)$$Z\left(f\right)=R_{sol}+\frac{R_{1}}{R_{1}{CPE}_{1}{\left(j2\pi f\right)}^{n_{1}}+1}+\frac{R_{2}}{R_{2}{CPE}_{2}{\left(j2\pi f\right)}^{n_{2}}+1}$$

In Table 5, it is noted that the corrosion resistance (R_p_ = R_1_ + R_2_) increased with the addition of silicon, reaching values up to 10^6^ Ω cm^2^ for the Ti15Mo7Zr0.5Si and Ti15Mo7Zr1Si samples, indicating that the materials present high corrosion resistance, being higher for Ti15Mo7Zr0.5Si.

### 3.3. Three-Point Bending Test

Table 6 shows the modulus of elasticity’s average values (E) of the four specimens tested with their respective standard deviation, which were found by following Equation (3), considering the applied load (F), the distance between supports at which the equipment’s lower shank is placed for placing samples (L), the moment of inertia (I), and the deformation (δ):(3)$$E=\frac{F\cdot L^{3}}{48\cdot \delta \cdot I}\cdot 10^{-3}$$

Since the specimens have a rectangular cross-section, Equation (4) defines the moment of inertia where w is the specimen’s width and h is its thickness.
(4)$$I=\frac{w\cdot h^{3}}{12}$$

After calculating the average values of each specimen’s elasticity modulus, it can be observed that the lower E value was obtained for the TiMo15Zr7Si0.5 sample (82.4 ± 9.2 GPa), while the upper E value was for TiMo15Zr7 (61.5 ± 8.4 GPa). This fact indicates that the small addition of silicon could help the samples to decrease their Young’s modulus and to approach the E of human bone (7–30 GPa).

Nevertheless, the results obtained are higher than those of the TiMoSi alloy (20–43 GPa) and within ranges that are comparable to those discovered in earlier studies, such as those of the TiMoZrTa alloy (52–69 GPa). The research alloy, TiMoZrSi, has a substantially lower Young’s modulus when compared to more widely used alloys in industry, medicine, and dentistry, such as stainless steel (190–210 GPa), cpTitanium (105 GPa), Ti6Al4V (110 GPa), and CoCrMo (210–253 GPa) [35,36].

### 3.4. Microhardness Test

The microhardness values in HV 0.005, HV 0.025, and HV 0.05, as well as the standard deviation and the minimum and maximum values for the 10 indentations made on each sample, are shown in Figure 8 and Table 7.

Figure 8 shows a comparison of the 10 indentations performed on the four specimens for each load applied and their calculated averages. In general, the values obtained range between 300 and 450 HV for the three applied loads, although some very high peaks can be observed against other very low ones, which would indicate the existence of harder or softer areas on the surface of the samples and, therefore, a slight lack of homogeneity during the manufacturing process of the samples due to the varying hardness ratings of the alpha and beta phases and the orientation of the material’s crystals. Additionally, for the three loads applied, the mean HV values of the studied samples were around 350 and 400 HV, approximately.

Table 7 shows more clearly the obtained averages, where for 5 gf applied, the TiMo15Zr7Si1 sample achieved a higher value (396 ± 32 HV), while for 25 and 50 gf, the sample that presented higher microhardness values was TiMo15Zr7Si0.75 (389 ± 13 HV; 395 ± 14). In turn, for the 5 gf load, the minimum and maximum values were 247 and 451 HV (TiMo15Zr7Si0.5), respectively. For the 25 gf loading, the minimum value was 268 HV, and a maximum of 433 HV was reached for the TiMo15Zr7Si1 sample. Finally, when applying the 50 gf load, TiMo15Zr7Si0.5 obtained the minimum and maximum values of 218 and 427 HV, respectively. Therefore, by adding silicon to the samples, a slight increase of the hardness values appears. 

Moreover, the values found in this research were lower than those of conventional alloys like CoCrMo (155–601 HV) and Ti6Al4V (541 HV) [35], but higher than those of cpTi [37,38].

By measuring hardness at different loads, we can observe how the hardness of the material changes with increasing plastic deformation. This information is especially important for understanding the alloy’s resistance to deformation, its ability to withstand applied loads, and its response to forming or shaping processes. In some cases, variations in hardness values with different loads may suggest the presence of localized variations in the microstructure, such as variations in grain size, phases, or the presence of inclusions. By examining the hardness through different loadings, the different phases of the material’s microstructure can be identified.

## 4. Conclusions

Four new titanium alloys’ mechanical properties and corrosion behavior have been compared and contrasted in this study. Microstructural, corrosion potential and corrosion rate curves, EIS, three-point bending, and microhardness results have produced the following findings:After the samples were exposed to Kroll reagent, biphasic dendritic patterns were seen on their surfaces. The dendrites became smaller when silicon was added because silicon is a strong grain refiner in titanium alloys. It promotes the formation of fine grains that hinder the propagation of cracks, enhancing the material’s resistance to fatigue.Both samples’ potential values grew during the immersion duration without diminishing, indicating that the passive layer, which was produced on the surface of each sample, is thermodynamically resilient under standard circumstances. Silicon oxide forms a protective oxide layer on the alloy surface, reducing the rate of oxidation and improving the alloys’ resistance to body fluid degradation.Since the polarization resistance was fairly large, the low corrosion currents and rates showed the tested samples’ exceptional performance in the evaluated environment, Ringer solution. R(QR)(QR) was the EC that most closely fit the measured EIS data, and as additional positive potentials were applied, the phase angle and the impedance results tended to rise. The sample 0.5%Si had the lowest corrosion rate because this is the solubility limit of silicon in the particular titanium alloy, and when exceeding this content, the presence of Si may create galvanic couples between the Si-containing regions and the rest of the titanium alloy, leading to localized electrochemical reactions.By examining the hardness through different loadings, two different phases, one hard and one soft, of the material’s microstructure were identified. Microhardness and elastic modulus values for both instances were lower than those for several conventional biomedical alloys.

In general, all samples showed good chemical and biological qualities, with slightly better results for the Ti15Mo7Zr0.5Si sample. 

## Figures and Tables

**Figure 1 materials-16-04906-f001:**
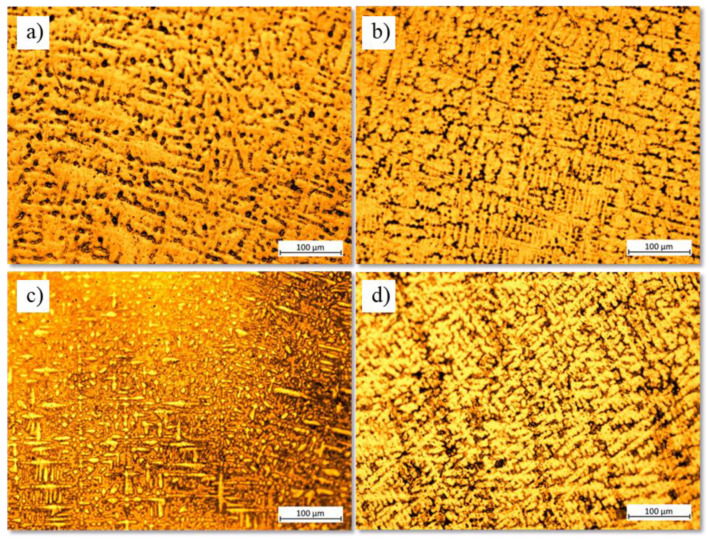
Optical microstructure after Kroll reagent etching for (**a**) Ti15Mo7Zr, (**b**) Ti15Mo7Zr0.5Si, (**c**) Ti15Mo7Zr0.75Si, and (**d**) Ti15Mo7Zr1Si samples.

**Figure 2 materials-16-04906-f002:**
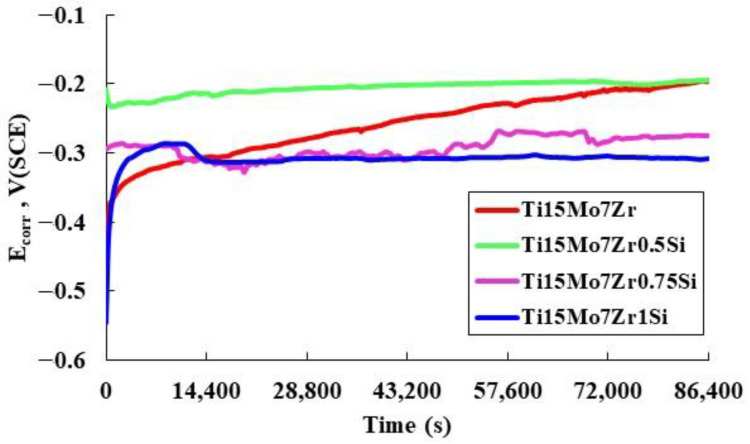
Corrosion potential curves for Ti15Mo7Zr, Ti15Mo7Zr0.5Si, Ti15Mo7Zr0.75Si, and Ti15Mo7Zr1Si alloys after 24 h’ immersion.

**Figure 3 materials-16-04906-f003:**
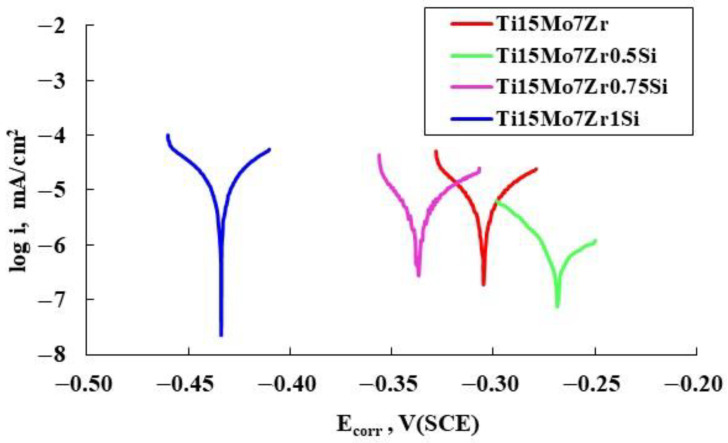
Polarization curves of the four test samples.

**Figure 4 materials-16-04906-f004:**
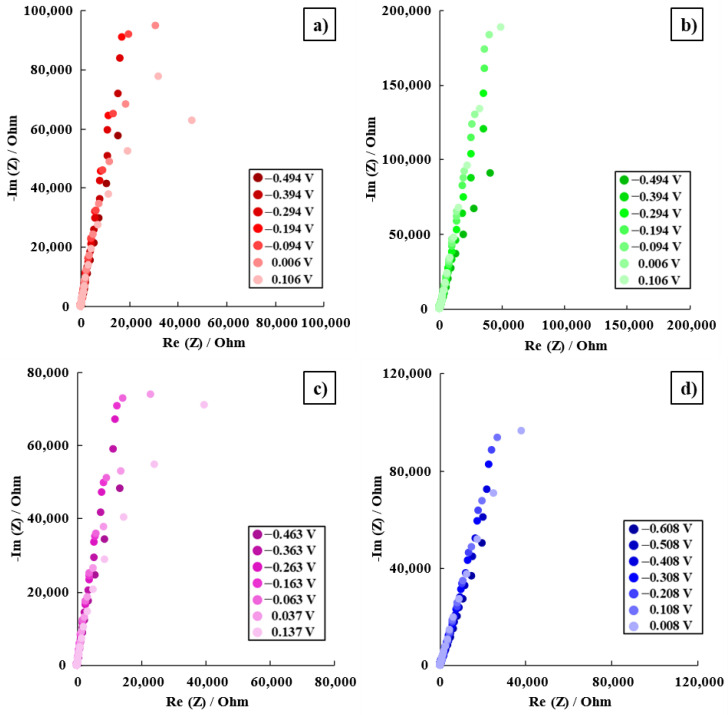
Nyquist diagrams for Ti15Mo7Zr (**a**), Ti15Mo7Zr0.5Si (**b**), Ti15Mo7Zr0.75Si (**c**), and Ti15Mo7Zr1Si (**d**) in Ringer’s solution.

**Figure 5 materials-16-04906-f005:**
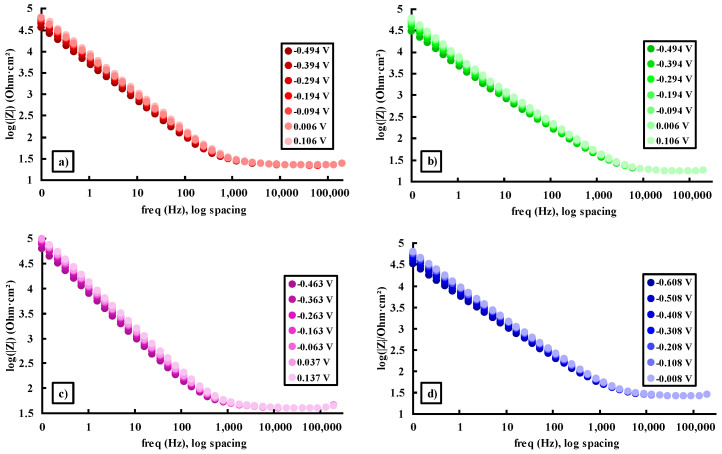
Bode impedance diagrams for Ti15Mo7Zr (**a**), Ti15Mo7Zr0.5Si (**b**), Ti15Mo7Zr0.75Si (**c**), and Ti15Mo7Zr1Si (**d**) in Ringer’s solution.

**Figure 6 materials-16-04906-f006:**
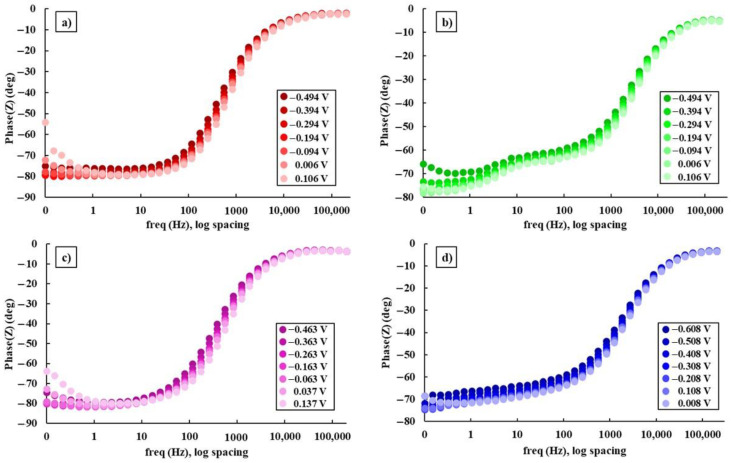
Bode phase graphs for: Ti15Mo7Zr (**a**), Ti15Mo7Zr0.5Si (**b**), Ti15Mo7Zr0.75Si (**c**), and Ti15Mo7Zr1Si (**d**) in Ringer’s solution.

**Figure 7 materials-16-04906-f007:**
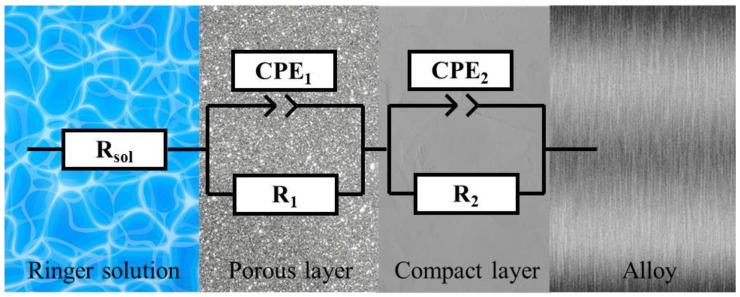
Equivalent circuit R(QR)(QR).

**Figure 8 materials-16-04906-f008:**
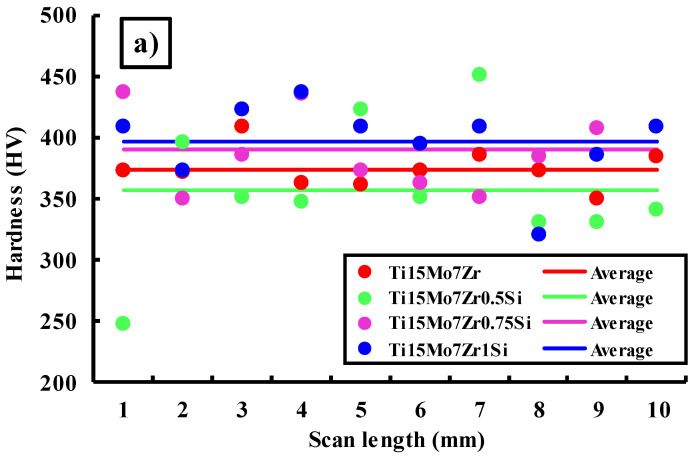
Microhardness values for each indentation for the four samples under study under loadings of 5 (**a**), 25 (**b**), and 50 (**c**) gf.

**Table 1 materials-16-04906-t001:** Ti15Mo7ZrxSi experimental samples’ chemical composition obtained by EDX (wt %).

Alloy	Ti (%)	Mo (%)	Zr (%)	Si (%)
Ti15Mo7Zr	77.87	15.03	7.10	-
Ti15Mo7Zr0.5Si	77.52	15.10	6.89	0.49
Ti15Mo7Zr0.75Si	78.15	14.59	6.53	0.73
Ti15Mo7Zr1Si	77.24	15.00	6.79	0.97

**Table 2 materials-16-04906-t002:** For the four samples dipped in Ringer Grifols solution, open-circuit potential (OCP) measurements were made initially, one hour later, and one day later.

Alloy	OCP, V vs. SCE		
	Initial	After 1 h	After 1 Day
TiMo15Zr7	−0.432	−0.337	−0.194
TiMo15Zr7Si0.5	−0.207	−0.227	−0.194
TiMo15Zr7Si0.75	−0.294	−0.284	−0.263
TiMo15Zr7Si1	−0.546	−0.307	−0.308

**Table 3 materials-16-04906-t003:** Corrosion parameters for all samples tested.

Parameters	Ti15Mo7Zr	Ti15Mo7Zr0.5Si	Ti15Mo7Zr0.75Si	Ti15Mo7Zr1Si
E_corr_ (mV vs. Ref)	−305.03	−264.59	−337.50	−445.46
I_corr_ (nA/cm^2^)	2.00	2.40	2.00	4.00
β_c_ (mV/dec)	13.30	14.70	12.00	9.60
β_a_ (mV/dec)	15.30	16.70	15.80	10.00
Equivalent weight (g/eq)	58.12	58.02	57.97	57.92
Density (g/cm^3^)	5.50	5.49	5.48	5.48
Surface (cm^2^)	0.62	0.31	1.23	0.63
Corrosion rate (μm/year)	0.11	0.03	0.06	0.22
Rp (Ω·cm^2^)	9.86 × 10^5^	6.09 × 10^6^	1.2 × 10^6^	4.48 × 10^5^

**Table 4 materials-16-04906-t004:** Results from the samples’ Bode diagrams under study at E_corr_ and at ±300 mV.

Samples	Potential (V)	Maximum Impedance (Ω)	Maximum Phase Angle (°)
	−0.494	59,795.00	76
Ti20Mo7Zr	−0.194	92,456.59	80
	−0.106	77,547.44	79
	−0.494	99,123.73	70
Ti20Mo7Zr0.5Si	−0.194	164,512.91	77
	−0.106	194,874.17	77
	−0.463	50,019.06	79
Ti20Mo7Zr0.75Si	−0.163	72,085.70	82
	−0.137	81,420.35	80
	−0.608	54,039.71	68
Ti20Mo7Zr1Si	−0.308	85,686.74	74
	−0.008	103,741.00	72

**Table 5 materials-16-04906-t005:** Equivalent circuit R(QR)(QR) of samples Ti15Mo7Zr0.5Si and Ti15Mo7Zr1Si.

Sample	Potential (V)	R_sol_ (Ω·cm^2^)	Y_01_ (S·s^n^/cm^2^)	n_1_	R_1_ (Ω·cm^2^)	Y_02_ (S·s^n^)	n_2_	R_2_ (Ω·cm^2^)	χ^2^
Ti15Mo7Zr	0.106	22.47	2.04 × 10^−5^	1	1.19	2.07 × 10^−5^	0.90	1.99 × 10^5^	6.55 × 10^−4^
Ti15Mo7Zr0.5Si	0.106	17.04	7.52 × 10^−5^	0.73	401.20	2.55 × 10^−5^	0.87	1.67 × 10^6^	6.86 × 10^−4^
Ti15Mo7Zr0.75Si	0.137	39.91	4.55 × 10^−4^	0.65	620.30	1.35 × 10^−5^	0.93	2.42 × 10^5^	9.63 × 10^−4^
TiMo15Zr71Si	−0.008	26.08	1.48 × 10^−4^	0.75	100.80	2.27 × 10^−5^	0.81	1.52 × 10^6^	4.44 × 10^−4^

**Table 6 materials-16-04906-t006:** Young’s modulus of the studied alloys.

Sample	E Average (GPa)
TiMo15Zr7	82.4 ± 9.2
TiMo15Zr7Si0.5	61.5 ± 8.4
TiMo15Zr7Si0.75	66.0 ± 8.0
TiMo15Zr7Si1	73.4 ± 8.2

**Table 7 materials-16-04906-t007:** Microhardness values of applied loads of the four samples for 5, 25, and 50 gf.

Load (gf)	Samples	Average (HV)	Min	Max
5	TiMo15Zr7	374 ± 16	349	408
	TiMo15Zr7Si0.5	357 ± 56	247	451
	TiMo15Zr7Si0.75	389 ± 32	350	437
	TiMo15Zr7Si1	396 ± 32	320	437
25	TiMo15Zr7	348 ± 21	317	392
	TiMo15Zr7Si0.5	365 ± 24	334	403
	TiMo15Zr7Si0.75	389 ± 13	366	409
	TiMo15Zr7Si1	382 ± 50	268	433
50	TiMo15Zr7	339 ± 32	258	372
	TiMo15Zr7Si0.5	362 ± 63	218	427
	TiMo15Zr7Si0.75	395 ± 14	379	419
	TiMo15Zr7Si1	382 ± 22	345	406

## Data Availability

Not applicable.

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
