# Peer review of "Effect of Si Contents on the Properties of Ti15Mo7ZrxSi Alloys"

_materials, 2023, doi:10.3390/ma16144906_

Round 1

Reviewer 1 Report

This paper was well prepared and organized from the point that it showed the effect of Si contents on the properties of Ti alloys for medical uses. But some part of the manuscript needs to be revised as follows. (When the authors respond the answer, please write to file including the below questions)

1. Title is to be revised to show the purpose of this manuscript. I recommend - "Effect of Si contents on the properties of Ti15Mo7ZrxSi allloys".

2. Keywords: Ti alloys are deleted.

3. Introduction: Why don't you refer your paper - "New titanium alloys, promising materials for medical devices", materials, 14, 5934(2021). Please rewirte the introduction including the effect of Si contents of the properties of Ti alloys.

4. Methods

  1) All of the instruments used needs the information about 'model, manufacrurer, country'.

  2) Table 1 MUST be rewritten. Show the analyzed composition including the impurities.

  3) Figure 1 must be deleted, because it is very basic.

  4) Revise the sub-title - Microstructure Observation

  5) Line 135; Ringer solution needs the related references. What is the unit of the constituents?

  6) Line 142; I don't understand 10V of potential when you measure corrosion potential. Please explain it. Test temperature? Deaeration or not?

  7) Line 145; Usually, corrosion rate is expressed as icorr. Test temperature? Deaeration or not?

  8) EIS; What is DC potential, AC amplitude, Frequency??? Please describe the test conditions precisely.

5. Results

  1) Figure 2; a little low magnification. Usually, (a) Ti15Mo7Zr etc...

  2) Figure 3; Vertical axis must be revised - Ecorr, V(SCE). What is the effect of Si contents on the corrosion potential? Title must contain the information about test solution, temperature, deaerated or not. Did you depassivate the surface of the specimen before measuring corrosion potentials? Because the Ringer solution is neutral, the depassivation treatment is very important to measure corrosion potential.

  3) Table 2; Please plot the diagram using the data of table and explain the effect of Si content on corrosion potential.

  4) Figure 4. Before you present EIS results, SHOW the polarization curves. Please replot! Anyway, Why did you change the measuring potential from anodic to cathodic?

  5) Table 2; Why did you change the measuring potential from anodic to cathodic?

  6) Figure 8; Replot the diagram. X is Evorr, V(SCE), y is log i, A/cm^2

  7) Replot the diagram combining Figure 9 and Table 7. Why did you change the load in hardness measuments?

6. Conclusions

  1) Based on the above points the conclusions need to be more deep and clearer on also the drawbacks of this process, it may be fine to generally promote this process, but the authors should provide also a comprehensive and objective list of conclusions with the good the bad and the neutral conclusions.

  2) What is your conclusion about the effect of Si content on the properties?

  3) What is your conclusion about the effect of potential in EIS?

  4) What is your conclusion about the effect of load in hardness?

Author Response

In the name of all authors of present manuscript, I express our gratitude for all provided comments. We hope that we have made all necessary corrections according to your suggestions.

Bellow, we present the answer for all observations/suggestions and the new references in the manuscript.

Author Response

Dear reviewer,

In the name of all authors of present manuscript, I express our gratitude for all provided comments. We hope that we have made all necessary corrections according to your suggestions.

Bellow, we present the answer for all observations/suggestions and the new references in the manuscript.

  1. Are they the same size? It is very important to ensure the same test area for evaluating the corrosión resistance of samples.

Answer: The samples have not the same size, we have measured it with Image J program and we have considered the obtained values in order to obtain the corrosion resistance.

  1. In Fig. 8 (Polarisation curves), is the ordinate unit mA or A?

Answer: We have changed it. Thank you very much.

  1. From the perspective of corrosión resistance, it can be found that the Sample Ti15Mo7Zr0.5Si shows the best corrosión resistance. Please explain why the corrosión resistance of titanium-based alloy first increases and then decreases with the increase of Si content.

Answer: If the added Silicon content exceeds 0.5%, the corrosion rate increased due to the formation of galvanic couples between the Si-containing regions and the rest of the titanium alloy, leading to localized electrochemical reactions. The Si content may influence the composition, thickness and stability of the passive oxide layer formed on the surface of the alloy. Changes in the passivation behavior can result in variations of the corrosion rate.

Round 2

Reviewer 1 Report

Line 153 : EIS test can be measured at the corrosion potential or any potentials. What is your test condition? (DC potential's meaning. Please recheck it)

Line 255: The unit of mpy is not standard. It must be changed as micrometer/year's unit.

Author Response

Dear reviewer,

In the name of all authors of present manuscript, I express our gratitude for all provided comments. We hope that we have made all necessary corrections according to your suggestions.

Bellow, we present the answer for all observations/suggestions and the new references in the manuscript.

Line 153 : EIS test can be measured at the corrosion potential or any potentials. What is your test condition? (DC potential's meaning. Please recheck it)

Answer: Thank you, we have added in the manuscript: " To analyze the characteristics of the oxide film, the impedance spectra were recorded at 7 different DC potentials around corrosion potential, in the range Ecorr ± 300 mV, with a 100 mV step from Ecorr, permitting the system to stabilized for 5 min at each potential."

Line 255: The unit of mpy is not standard. It must be changed as micrometer/year's unit.

Answer: Thank you, we have changed it in the table and in the text.
